# Ultra-Efficient Removal of Crystal Violet Dye Using Industrial Brine and Horn-Derived Biochar: Synergistic Action of Salting-Out/Adsorption

**DOI:** 10.3390/toxics13121039

**Published:** 2025-11-30

**Authors:** Asma Nouioua, Dhirar Ben Salem, Abdelkader Ouakouak, Saadia Guergazi, Abdelouaheb Abdelli, Daniel Goma, Jose Manuel Gatica, Hilario Vidal

**Affiliations:** 1Research Laboratory in Subterranean and Surface Hydraulics, University of Biskra, P.O. Box 145, Biskra 07000, Algeria; dhirar.bensalem@univ-biskra.dz (D.B.S.); s.guergazi@univ-biskra.dz (S.G.); 2Department of Industrial Chemistry, Faculty of Science and Technology, University of Biskra, P.O. Box 145, Biskra 07000, Algeria; 3Hydraulic and Civil Engineering Department, University of El Oued, P.O. Box 789, El Oued 39000, Algeria; ouakouakk@yahoo.fr; 4Civil Engineering and Hydraulic Department, University of Biskra, P.O. Box 145, Biskra 07000, Algeria; 5Department of Quality Control, ENASEL Unit, El-Outaya Complex, Biskra 07030, Algeria; abdelouahebabdelli2@gmail.com; 6Departamento de Ciencia de los Materiales e Ingeniería Metalúrgica y Química Inorgánica, Instituto de Microscopía Electrónica y Materiales (IMEYMAT), Universidad de Cádiz, 11510 Puerto Real, Spain; dani.gomajimenez@gm.uca.es (D.G.); hilario.vidal@uca.es (H.V.); 7School of Chemistry and Chemical Engineering, Queen’s University Belfast, David-Keir Building, Strandmillis Road, Belfast BT9 5AG, UK

**Keywords:** brine discharge, biochar, crystal violet removal, mechanisms, cost analysis

## Abstract

This study introduces an innovative hybrid approach combining salting-out and adsorption for the highly efficient removal of crystal violet (CV) dye from aqueous solutions. The method leverages high-ionic-strength brine discharge from the Complex of El-Outaya (CEO, ENASEL, Biskra, Algeria) and micro-mesoporous biochar derived from calves’ horn cores (BHC-800). Results demonstrate that both undiluted and diluted brine significantly enhance CV removal, while BHC-800, with a surface area of 258 m^2^ g^−1^, exhibits a maximum Langmuir adsorption capacity of 106.1 mg g^−1^ (at 20 °C ± 2). Thermodynamic analysis confirms a spontaneous (Δ*G°* < 0) and exothermic (Δ*H°* = −0.86 kJ mol^−1^) process, with increased interfacial disorder (Δ*S°* = 93.53 J mol^−1^ K^−1^). The synergistic effect of salting-out and adsorption achieved ~99.8% removal of CV at an initial concentration of 1000 mg L^−1^. Furthermore, BHC-800 exhibited excellent reusability, maintaining high adsorption efficiency over multiple cycles. Economic assessment revealed operational costs of 0.45–0.89 US$ m^−3^ for 60% brine discharge. Biochar production costs were 0.076–0.18 US$ kg^−1^, translating to 7.5–17.2 (10^−4^ US$) per gram of CV removed. This dual strategy not only offers an eco-friendly and cost-effective solution for dye-laden water but also promotes the valorization of saline effluents and animal byproducts, addressing critical environmental challenges in industrial wastewater treatment.

## 1. Introduction

Water pollution is a critical environmental issue, worsened by industrial effluents contaminating water resources. Among these pollutants, crystal violet (CV), a toxic cationic dye widely used in textiles, printing, and cosmetics, poses significant ecological and health risks [1]. Improper disposal of CV can cause eye irritation, nausea, cancer, renal failure, genetic mutations, and reproductive defects, emphasizing the need for stringent wastewater management measures [2]. As CV is resistant to biodegradation, advanced remediation technologies are essential to mitigate its impact.

Various industries, including salt production, sodium hypochlorite manufacturing, desalination plants, oil and gas drilling, mining, and others, generate significant volumes of saline effluents and concentrated brine [3]. These by-products, often rich in salts, are typically treated as waste, leading to environmental management challenges [4]. However, valorizing such effluents for salting-out processes aligns with sustainability and circular economy principles, offering significant ecological and economic benefits [5]. Introducing high salt concentrations into aqueous solutions reduces the solubility of specific contaminants and leads to their precipitation or separation [6,7]. This technology, known as the salting-out process, has gained attention as a promising physicochemical technique for pollutant separation, particularly for dyes with limited solubility in water [8].

In addition to the salting-out, the adsorption process has emerged as one of the most efficient and straightforward techniques for pollutant removal by applying different adsorbent materials such as zeolite, clay, and carbonaceous [9,10]. Furthermore, using waste biomaterials as adsorbents not only addresses waste disposal challenges but also provides an eco-friendly alternative to synthetic and costly adsorbents like activated char [11].

Animal bones, a significant by-product of the food industry, generate approximately 130 billion kg of waste annually [12]. Comprising 18% of an animal’s live weight, these residues are rich in calcium and phosphorus oxides and are often landfilled or repurposed for animal feed and fertilizers [13]. When heated, they yield valuable compounds such as beta-tricalcium phosphate and hydroxyapatite, making them effective catalysts in various applications [12]. According to the literature, biochar derived from animal by-products has been reported to be an effective material for dye-contaminated water treatment, owing to its high porosity and large surface area [13].

To date, the literature is dominated by studies on individual adsorption techniques for organic dye removal using various adsorbents. However, the integration of a salting-out process—utilizing brine effluents as a low-cost, valorized resource—with adsorption using locally sourced biochar derived from animal bones has not yet been explored. This research gap motivated our group to investigate a zero-waste approach for effective dye degradation. Accordingly, this study aims to utilize both brine effluent and biochar derived from calves’ horns as a novel, green, and low-cost method for removing crystal violet dye. The research investigates CV dye removal through salting-out and adsorption processes, both individually and in combination, to maximize removal efficiency and minimize operational costs. The performance of brine and biochar in dye removal was experimentally evaluated under various working conditions (pH, concentration, dosage, contact time, and temperature), supported by the characterizations of materials and numerical modeling using kinetic and isotherm models. By exploring the synergy between salting-out and adsorption, the study provides an innovative, sustainable solution for wastewater treatment, promoting the reuse of saline effluents and animal-derived waste in environmental management.

## 2. Materials and Methods

In this study, Crystal Violet (CV), also known as Gentian Violet or Basic Violet 3 (C_25_H_30_ClN_3_, MW 407.99 g mol^−1^), was used with a purity of 99% (supplied by BIOCHEM Chemopharma, Laval, QC, Canada). It has a density of 1.19 g cm^−3^, solubility of 16 g L^−1^ in water at 25 °C, and identifiers CAS 548-62-9, EINECS 208-953-6, and CI 42555. All solutions were prepared with deionized water, using analytical-grade chemicals.

### 2.1. Brine Discharge and BHC-800 Biochar

#### 2.1.1. Brine Discharge

The saline solutions used in the salting-out experiments were sourced from brine effluents generated during the salt-washing process at the Salt Complex of El-Outaya (CEO) in Biskra, Algeria (35°01′44.6″ N 5°36′06.2″ E), which is affiliated with the National Salt Company (ENASEL) (Appendix A). This facility processes raw salt rock extracted from the El M’ghair region in southern Algeria. The salt deposit is geologically characterized by massive halite layers interspersed with clay and gypsum [14].

#### 2.1.2. Production of Adsorbent BHC-800 Biochar

The preparation of biochar from the inner bony core of calves’ horns began with fresh horns obtained from a local abattoir in Biskra, Algeria (34°51′20.2″ N 5°46′11.4″ E). To preserve freshness, the horns were wrapped in plastic film and stored at 4 °C. Each horn’s keratin layer was softened by soaking in hot water for 1 h, enabling its removal. The dense inner core was then washed thoroughly with distilled water until clean, cut into small pieces, and oven-dried at 105 °C for 24 h (POL-EKO SLW 115 STD, Wodzisław Śląski, Poland).

The dried material was ground, sieved (0.5–1 mm), and carbonized in a muffle furnace (DAIHAN Scientific, Wonju, Republic of Korea) at 800 °C for 3 h under limited oxygen to produce biochar (Appendix A).

After cooling under inert conditions, the biochar was sieved into 75–250 µm, treated with 0.1 M HCl for 10 min, filtered, and rinsed with distilled water until the pH stabilizes between 6.0 and 7.0. The purified biochar was dried again at 105 °C for 24 h, labeled as BHC-800, and stored in airtight containers for future use.

### 2.2. Characterization of Brine Discharge and BHC-800 Biochar

The mineral composition of raw salt and the physicochemical properties of brine discharge were analyzed at the ENASEL-CEO facility (Biskra, Algeria) to ensure experimental consistency. The BHC-800 biochar was characterized using several techniques. X-ray Diffraction (XRD) analysis was conducted using a Bruker D8 Advance diffractometer (Bruker AXS, Karlsruhe, Germany) at a scanning rate of 10° min^−1^ to study the crystalline structure of the biochar. FTIR analysis was performed using a SHIMADZU-8400S spectrometer (Shimadzu Corporation, Kyoto, Japan) to identify surface functional groups before and after dye adsorption. Spectra were recorded over the range of 400–4000 cm^−1^ with a resolution of 2 cm^−1^. For analysis, pellets were prepared by pressing 1 mg of BHC-800 with 200 mg of KBr. Scanning Electron Microscopy (SEM) and Energy Dispersive X-ray (EDX) analysis were conducted using a FEG Nova NanoSEM 450 microscope (FEI Company, Hillsboro, OR, USA), providing insights into surface morphology and elemental composition. Brunauer–Emmett–Teller (BET) surface area analysis was performed with an Autosorb iQ3 instrument (Quantachrome Instruments, Boynton Beach, FL, USA), using N_2_ adsorption–desorption at −196 °C testing to measure surface area and pore structure. The samples were outgassed at 180 °C for 10 h prior to this analysis. Thermogravimetric analysis (TGA) was performed with a TA-Q50 instrument (New Castle, DE, USA), covering 30 °C to 900 °C (10 °C min^−1^) in an N_2_ atmosphere. The point of zero charge pH_PZC_ was determined by the drift method using a HANNA HI 2210 pH meter (Hanna Instruments, Bucharest, Romania), as detailed in previous studies [15].

### 2.3. Experiments

#### 2.3.1. Salting-Out and Adsorption Processes

The batch salting-out and adsorption experiments were conducted to assess the removal efficiency of CV dye under controlled conditions. For the salting-out experiments, different concentrations of saline solutions, derived from brine effluents (pH_solution_ ≈ 7.2), were mixed with crystal violet solutions in a 1:1 liquid–liquid ratio, resulting in initial concentrations ranging from 25 to 1000 mg L^−1^. The mixtures were stirred at 450 rpm for 20 min, then left to settle for 24 h at a temperature of 15 (±2) °C. The adsorption studies focused on the performance of BHC-800, with experiments conducted using 25 mL dye solutions (pH of distilled water = 6.6–7.0), being the sample dried at 105 °C before each test.

The effect of key parameters on CV removal, including pH (3–11), ionic strength (0.5–1.0 M NaCl), adsorbent dosage (0.5, 1, and 2 g L^−1^), and temperatures (20, 40, and 60 ± 2 °C), was investigated. The influence of pH on salting-out and adsorption efficiency was examined by adjusting pH using 0.1 M solutions of either NaOH or HCl. Kinetic studies were conducted with dye concentrations ranging from 5 to 600 mg L^−1^ and contact times from 0 to 360 min. The temperature difference between the salting-out (15 ± 2 °C) and adsorption (20 ± 2 °C) experiments reflects the local laboratory temperatures in Biskra, an arid region with strong seasonal fluctuations. The salting-out tests were conducted during a cooler period, when the ambient indoor temperature was around 14–17 °C. The adsorption experiments took place later, when ambient temperatures rose to 19–22 °C. The tests performed out of ambient conditions were thermally controlled (40 and 60 °C).

Chemical regeneration of adsorbents often uses acidic or basic solutions, which induce the release of adsorbed molecules. In this study, after reusing biochar for several adsorption cycles, the CV-loaded biochar was regenerated using 0.1 M HCl at a solid–liquid ratio of 1 g per 50 mL. The mixture was stirred at 150 rpm for 2 h at 20 ± 2 °C, then the biochar was washed to neutral pH, dried at 105 °C for 12 h, and reused in the subsequent adsorption cycle.

After treatment, mixtures were filtered (45 μm), and residual dye concentrations were measured spectrophotometrically by SP-3000 Plus spectrophotometer (Optima Inc., Tokyo, Japan) at 581 nm. Calibration curves ensured data precision. To validate results, blank tests and triplicates were performed, and rigorous protocols were applied for storing, cleaning, and handling the adsorbent, solutions, and containers.

After individually optimizing the operating conditions for both processes to determine the best removal conditions for CV, the combination was carried out in a sequential mode (Appendix A). In the first stage, salting-out was employed to induce phase separation of CV using saline effluents over 24 h. After that, the solution underwent a second-stage adsorption with BHC-800 to reduce or complete remove residual dye concentration. This cascade approach leverages the efficiency of salting-out in concentrating the dye, followed by adsorption to achieve maximum removal.

#### 2.3.2. Data Processing and Calculation

Appendix A provides an overview of the equations and methodologies employed in this comprehensive analysis. The removal efficiency of CV dye was assessed through analytical calculations, while data modeling and goodness-of-fit (*R*^2^ and SD) were evaluated using advanced kinetic models, including pseudo-first-order (PFO), pseudo-second-order (PSO), Avrami, and Elovich models. Adsorption behavior was further characterized using isotherm models such as Langmuir, Freundlich, and Temkin. Additionally, thermodynamic parameters (Δ*G°*, Δ*H°*, and Δ*S°*) were determined to elucidate the energy changes, spontaneity, and molecular interactions governing the adsorption process.

## 3. Results and Discussion

### 3.1. Salting-Out Study

#### 3.1.1. Properties of Raw Salt and Brine Discharge

The physicochemical characterization of raw salt and brine discharge (Table 1) is essential for evaluating their role in salting-out mechanisms and ensuring the efficacy of experimental procedures. The raw salt, with a NaCl content of 96.6% and minimal impurities such as magnesium chloride, calcium, and magnesium sulfates, offers a high-purity ionic medium for crystal violet retention. Its low insoluble residue (0.16%) and acceptable moisture content (3.23%) contribute to its stability during handling. The brine discharge, with a density of 22 °Bé and NaCl concentration of 175.43 g L^−1^, exhibits significant salinity and ionic strength. Chloride ions (157.4 g L^−1^) dominate, while magnesium (8.8 g L^−1^) and calcium (0.8 g L^−1^) influence electrostatic interactions with the dye. Additional ions such as sulfate and bicarbonate enhance the salting-out effect, reinforcing the brine’s effectiveness in dye retention.

#### 3.1.2. Effect of Brine Discharge Concentration

The effect of initial brine discharge concentration (*C*_i-BD_) on the retention of CV dye for an initial concentration of 25 mg L^−1^ can be understood through the salting-out mechanism, a critical process influencing dye precipitation and removal efficiency [16]. As depicted in Figure 1a,b, increasing the brine discharge concentration from 20% to 100% significantly improved removal efficiency, rising from 41.7% to 89.1%, for *C*_i-BD_ from 20% to 60%, and nearly complete removal of 99.8% at *C*_i-BD_ of 80%, and 100%. This indicates a progressive disruption of solvation shells and increased aggregation of CV molecules. At high salt concentrations, the increased ionic strength reduces the interaction between dye molecules and water, leading to further decreased solubility and promoting aggregation [17].

#### 3.1.3. Effect of Dye Concentration

The removal efficiency of CV via the salting-out process was evaluated over a CV concentration range of 100 to 1000 mg L^−1^ and at initial brine concentrations of 60% and 100%. The results indicate consistently high removal efficiency across all concentrations (Figure 1c). At 60% brine, removal efficiency declined from 90.3% at 100 mg L^−1^ to 81.5% at 1000 mg L^−1^ reflecting a decrease of approximately 8.8% compared to the lowest concentration (Figure 1c). This suggests that at lower CV concentrations, the salting-out effect is more effective due to the substantial disruption of solvation shells and enhanced dye aggregation. However, as *C*_o_ increases, competition among dye molecules reduces the efficiency of precipitation, leading to a gradual decline in removal. At 100% brine, removal efficiency remained consistently high, exceeding 99.2% across all CV concentrations. This near-complete removal indicates that at maximum ionic strength, solubility is drastically reduced, ensuring extensive CV aggregation and precipitation regardless of the initial concentration. The minimal decline from 99.5% to 99.21% for 100 mg L^−1^ and 1000 mg L^−1^, respectively, suggests that beyond a critical salt concentration, further increases in *C*_o_ have a negligible impact on removal efficiency.

#### 3.1.4. Effect of pH of the Medium

The effect of pH on the removal efficiency (R%) of crystal violet via the salting-out process was investigated using an initial CV concentration (*C*_o_) of 50 mg L^−1^ and varying initial brine discharge concentrations (*C*_i-BD_). Based on the results illustrated in Figure 1d, R% increased progressively with pH, highlighting the role of ionic interactions and the salting-out mechanism in modulating dye solubility and retention. At *C*_i-BD_ of 100%, R% ranged from 96.3% at pH 3.0 to 99.9% at pH 11.0, indicating a strong salting-out effect even under acidic conditions. Higher pH further enhanced efficiency due to reduced solubility of deprotonated CV molecules and increased ionic strength. At *C*_i-BD_ of 60%, R% was lower at acidic pH (75.6% at pH 3.0) but improved significantly with increasing pH, reaching 98.6% at pH 11.0. The reduced BD concentration limited ionic strength, making the process more pH-dependent. This facilitated the reaction between CV and hydroxide ions, leading to the complete conversion of CV into a new compound, solvent violet 9 [18]. A sharp improvement in R% was observed between pH 7.0 and pH 9.0, demonstrating the compensatory effect of higher pH in enhancing dye retention. Therefore, the salting-out process for CV retention is strongly influenced by pH, with higher retention efficiencies consistently observed under alkaline conditions.

### 3.2. Characterization of BHC-800 and Its Efficiency in CV Dye Adsorption

#### 3.2.1. XRD Analysis

Figure 2a displays the X-ray diffraction (XRD) pattern of BHC-800. The pattern exhibits several distinct crystalline peaks, the most intense of which appear at approximately 31.8° (211), 32.8° (112), and 34.0° (300) 2θ, forming the characteristic triplet of hydroxyapatite (Ca_10_(PO_4_)_6_(OH)_2_, HA) and partially carbonated hydroxyapatite (10.8° (001)) (JCPDS card No. 09-0432), the dominant mineral phase in bone [19,20,21,22,23]. Additional HA reflections are visible at around 25.8° (002), 39.7° (310), 46.7° (222), 49.5° (213), and 53.1° (004), further confirming the presence of a well-crystallized apatite structure after pyrolysis at 800 °C. A broad hump centered near 22–26° and peak at ~43.8 reflect the contribution of amorphous carbon from the biochar matrix [19,20]. The crystallite size of BHC-800 was estimated from the instrument-corrected FWHM values of the (002), (211), (112), (300), and (310) reflections using Scherrer’s equation in the HighScore Plus V. 3.0e (3.0.5) software (PANalytical, 2012, Almelo, The Netherlands). The calculated values yielded an average crystallite size of ~43 nm. Thus, the material consists mainly of crystalline hydroxyapatite embedded within an amorphous carbon matrix, a structure favorable for adsorption due to the coexistence of stable Ca–P mineral domains and porous carbon.

#### 3.2.2. FTIR Spectra Analysis

FTIR spectra of BHC-800 before and after adsorption of CV dye are shown in Figure 2b. The broad –OH stretching at 3300 cm^−1^ shows a change in intensity post-adsorption, indicating possible hydrogen bonding with hydroxyl groups on the material surface, a mechanism supported by similar findings in dye adsorption studies [24]. The peak at 2930 cm^−1^, commonly associated with aliphatic C-H groups, shows slight changes that may reflect weak van der Waals forces between the adsorbent and the dye, consistent with prior observations [25]. The peak at 2012 cm^−1^ likely arises from the thermal degradation of bone proteins, leading to the formation of nitrogen- and sulfur-containing groups such as isocyanate (–NCO), thiocyanate (SCN−), and isothiocyanate (–N=C=S) [26]. The aromatic C=C/C=O stretching at around 1600 cm^−1^ could be an indicator for the π-π interactions between the aromatic rings of BHC-800 and crystal violet, a well-documented mechanism in aromatic dye adsorption [27,28]. The strong band observed at around 1400 cm^−1^ may correspond to the C–C and C–N stretching [29]. The peak at 1210 cm^−1^ corresponded to the stretching vibrations of aromatic C−O [30]. Phosphate groups are represented by P–O bending vibration bands observed at 1060, 870, 600–740, and 560 cm^−1^ [28,31]. C–H bond is also explored at around 560 cm^−1^ [24]. Moreover, after the adsorption process, the infrared spectrum of BHC-800 reveals new bands at 919 cm^−1^ and 1105 cm^−1^. These bands could be attributed to the C–O–C asymmetrical stretch, P–N stretching vibrations, or the C–N vibration characteristic of CV dye [32], which may support the fixation of the dye to the adsorbent.

#### 3.2.3. SEM-EDX Analysis

SEM images reveal the morphology of BHC-800 (Figure 3), highlighting its highly textured surface, characterized by significant roughness and irregular protrusions. The pyrolysis process used during the preparation of the adsorbent further enhances the material porosity. Additionally, the observed variation in particle size indicates a heterogeneous distribution, which is critical for increasing the surface area, essential for optimal adsorption performance.

According to the literature, different adsorbent materials containing carbon, oxygen, calcium, and phosphate have demonstrated effective dye adsorption performance in water [13,33,34]. In this context, EDX analysis (Figure 3b) of BHC-800 indicates the presence of 6.5% carbon, a high oxygen content (37.4%), calcium (37.5%), and phosphorus (17.3%). These findings suggest that BHC-800 is enriched with various oxygen-containing functional groups, which is consistent with the results obtained from FTIR analysis.

#### 3.2.4. N_2_ Physisorption Isotherm

N_2_ physisorption at −196 °C, analyzed via the BET and BJH methods, is a key technique for assessing the surface area and porosity of adsorbents—critical factors in pollutant removal [35]. Preparation methods, including pyrolysis and activation, influence these textural properties, directly impacting adsorption efficiency. According to Figure 4a, the BET N_2_ adsorption–desorption isotherm of BHC-800 biochar exhibits a Type IV isotherm with an H3-type hysteresis loop (or possibly H4), indicating the presence of mesopores associated with slit-shaped structures. This interpretation is further supported by the pore size distribution (Figure 4b), which shows dominant pore widths at ~1.9 nm (borderline microporous/mesoporous) and ~5.7 nm (mesoporous), confirming the micro–mesoporous nature of the material.

The mesoporosity of BHC-800 may facilitate dye diffusion to active sites, thereby enhancing adsorption. Its total pore volume (*V*_T_ = 0.186 cm^3^ g^−1^) and specific surface area (*S*_BET_ = 258 m^2^ g^−1^) suggest a high adsorption potential for crystal violet.

#### 3.2.5. TGA Analysis

Thermal stability and decomposition behavior of biochar BHC-800 were investigated under inert (N_2_) and oxidative (air) atmospheres using thermogravimetric analysis (TGA), as shown in Figure 5. The TGA curves reveal distinct degradation profiles depending on the atmospheric conditions. In the initial temperature range of 30–150 °C, a slight mass loss (~1–2%) was observed under both atmospheres, corresponding to the physically adsorbed moisture, indicating hydrophilic groups like hydroxyl (–OH) [36]. Between 150 and 400 °C, the decomposition of thermally unstable organic moieties and oxygenated functional groups—such as carboxylic acids, hydroxyls, and carbonyls—occurred [37]. This stage was more pronounced under air flow due to the onset of oxidative degradation. A significant difference was observed in the 400–700 °C range: under nitrogen, the mass loss was gradual, attributed to the slow pyrolytic degradation of aromatic structures and the progressive aromatization of the biochar matrix, whereas in air, a sharp decline in mass indicated vigorous oxidation and combustion of the fixed carbon content. At higher temperatures (700–900 °C), the weight stabilized under both conditions, representing the thermally stable inorganic residue. Specifically, the high residual mass ~87% at ~800 °C highlights the mineral content of the material, primarily calcium phosphate (hydroxyapatite) and other mineral phases [38]. However, the final residue, regardless of the ambient nature, reflects the mineral ash content, composed mainly of calcium phosphate and bone-related minerals. These results indicate the thermal stability of BHC-800 even under air and its high inorganic content, in accordance with XRD results (Section 3.2.1), suggesting its suitability for high-temperature applications.

#### 3.2.6. pH Value and Ionic Strength

The adsorption of CV dye by BHC-800 is influenced by pH, the pH_PZC_ of the adsorbent, and ionic strength. As a basic dye, CV exists as positively charged ions, and its adsorption behavior depends on the surface charge of BHC-800. Data obtained (Figure 6b) show a significant increase in adsorption capacity between pH 3.0 (*q*_e_ = 42.7 mg g^−1^) and 11.0 (*q*_e_ = 99.9 mg g^−1^) at an initial concentration (*C*_o_) of 100 mg L^−1^. Near the pH_PZC_ of BHC-800 (7.6, Figure 6a), the surface is neutral. Below this pH, protonation of surface species induces positive charges, causing electrostatic repulsion with cationic CV, reducing adsorption. At pH > pH_PZC_, the surface becomes negatively charged, enhancing electrostatic attraction and dye retention, with the mechanism being more pronounced at higher pH values.

Ionic strength has a minimal effect on CV adsorption (Figure 6c). An increase in NaCl concentration (0.05 to 1 M) leads to a slight decrease in adsorption capacity from 47.6 mg g^−1^ to 46.4 mg g^−1^ (*C*_o_ = 50 mg L^−1^), representing a 2.3% reduction. This is attributed to Na^+^ ions partially occupying adsorption sites, slightly reducing electrostatic attraction. However, the impact is minor, indicating that BHC-800 is resistant to variations in salt concentration.

Overall, while pH significantly enhances adsorption by altering surface charge, ionic strength plays a negligible role, confirming the stability of BHC-800 for CV removal under varying salt conditions. These findings align with similar studies [39].

#### 3.2.7. Effect of BHC-800 Dose and Contact Time on CV Adsorption

The effect of adsorbent dose on CV adsorption reveals distinct trends in adsorption capacity (*q*_e_) and removal efficiency (R%). As the dose increases from 0.5 to 2 g L^−1^ (Figure 7a), the *q*_e_ value decreases significantly across all initial concentrations (*C*_o_). For example, at *C*_o_ = 200 mg L^−1^, *q*_e_ drops from 185.6 mg g^−1^ at 0.5 g L^−1^ to 107.4 mg g^−1^ at 1 g L^−1^ and further to 69.3 mg g^−1^ at 2 g L^−1^. This decline is due to the reduced dye-to-adsorbent ratio at higher doses, which lowers the driving force for CV migration to the adsorbent surface. Additionally, higher dosages may cause particle clustering and agglomeration, hindering adsorption efficiency [40]. Conversely, R% increases with higher adsorbent doses, especially at low *C*_o_. For *C*_o_ = 40 mg L^−1^, R% rises from 85.7% at 0.5 g L^−1^ to 98.4% at 1 g L^−1^ and 99.7% at 2 g L^−1^, reflecting improved availability of active sites for dye removal [41]. At higher concentrations, however, R% diminishes at lower doses due to surface saturation at elevated pollutant loads, reducing the adsorbent efficiency. Accordingly, an adsorbent dose of 1 g L^−1^ was considered the efficient and cost-effective option.

Contact time significantly affects the adsorption process, particularly equilibrium uptake (Figure 7b). For BHC-800, adsorption capacity (*q*_t_) for CV at an initial concentration of 50 mg L^−1^ increases rapidly in the initial phase, reaching 17.1 mg g^−1^ at 5 min due to abundant active sites and a strong concentration gradient [42]. Between 10 and 120 min, *q*_t_ gradually rises from 17.9 mg g^−1^ to 29.6 mg g^−1^, reflecting progressive site saturation. Beyond 120 min, adsorption stabilizes around 47.5 mg g^−1^, indicating equilibrium as active sites become fully occupied and the driving force diminishes [43].

For kinetic modeling, based on Table 2, the PFO model showed poor correlation (*R*^2^ = 0.65 and SD = 8.03) and significant deviation between predicted (*q*_e_ = 43.3 mg g^−1^ ± 4.1) and experimental values, indicating that the simple physical adsorption is not dominant. The PSO model provided a better fit than the PFO (*R*^2^ = 0.77 and SD = 0.66), with a predicted *q*_e_ (48.1 mg g^−1^ ± 4.8) closely matching the experimental data. Generally, chemisorption-type interactions (electron sharing or exchange) are described by the PSO model. However, the Avrami and Elovich models offered the best fit (*R*^2^ = 0.88 and SD = 4.7–4.83). Avrami indicated a complex, multi-step mechanism involving both surface processes, diffusion steps, and chemisorption [44]. The predicted Avrami *q*_e_ value (49.2 mg g^−1^ ± 2.41) is comparable to that obtained from the PSO model, indicating that the Avrami model is also suitable for describing the adsorption capacities observed in the experimental data. The Elovich model supported chemisorption on a heterogeneous surface, showing a decreasing adsorption rate with increasing surface coverage [45]. To sum-up, the models suggest a dual mechanism in which initial rapid uptake occurs via high-energy surface reactions (chemisorption), followed by a slower stage controlled by surface/intraparticle diffusion, and site heterogeneity, consistent with the Avrami-driven multi-step behavior and Elovich.

#### 3.2.8. Adsorption Isotherm and Thermodynamic Studies

The adsorption of CV dye onto BHC-800 was evaluated across initial concentrations (*C*_o_) of 10–600 mg L^−1^. At low *C*_o_ (10–100 mg L^−1^), the removal efficiency (R%) (calculated from its equation in Appendix A) was high (73.3–99.4%) due to abundant active sites. However, R% declined to higher *C*_o_ values, reaching 33.8% at 600 mg L^−1^ due to site saturation. Adsorption capacity (*q*_e_) increased with *C*_o_, from 9.9 mg g^−1^ at 10 mg L^−1^ to 202.6 mg g^−1^ at 600 mg L^−1^, driven by enhanced mass transfer forces [39]. The effect of increasing temperature on CV dye adsorption was found to be negative, with a dramatic decline in adsorption capacity (*q*_e_).

For isotherm modeling as shown in Figure 7c and Table 3, the Langmuir adsorption capacities (*Q*_max_) decreased with increasing temperature: 106.1 mg g^−1^ at 20 °C, 97.9 mg g^−1^ at 40 °C, and 88.9 mg g^−1^ at 60 °C. The Langmuir model demonstrated a moderate fit with *R*^2^ = 0.68–0.71 and relatively high SD values, suggesting monolayer adsorption does not fully describe the system. Conversely, the Freundlich model provided the best fit with *R*^2^ = 0.96–0.99 and SD values = 3.10–5.34, indicating favorable multilayer adsorption on heterogeneous surfaces (*n*_F_ > 1). The Temkin model also showed strong performance with *R*^2^ = 0.90–0.96 and accompanied by decreasing SD values, reflecting adsorption on energetically diverse surface sites. These findings are consistent with prior research on the elimination of crystal violet [46,47,48].

The decrease in *Q*_max_ values with increasing temperature of CV solutions indicates reduced adsorption affinity at elevated temperatures. In essence, thermodynamic calculations presented in Table 3 show that negative Gibbs free energy values (Δ*G°*) confirmed the spontaneity of the process, while the exothermic nature of adsorption was supported by Δ*H*° = −0.86 kJ mol^−1^, consistent with weak adsorption such as van der Waals forces, electrostatic interactions, or hydrogen bonding [49]. A positive entropy change (Δ*S°* = 93.53 J mol^−1^ K^−1^) suggested increased randomness at the solid–liquid interface due to water displacement and redistribution of CV molecules, further confirming the strong interaction between CV and BHC-800.

For comparison, the adsorption capacities of BHC-800 are consistent with those reported in the literature (Table 4). Biochars derived from various plant and animal sources, including palm kernel shell, chinar leaf, Rumex acetosella leaves, and Durio zibethinus seed [50,51,52,53,54,55,56,57,58,59], have been widely studied for dye removal. The inner bony core of calves’ horns biochar exhibited high adsorption capacities (*Q*_max_ = 88.9–106.1 mg g^−1^) at pH 6.5 and 1 g L^−1^, surpassing many plant-derived biochars. However, each biochar reported in the literature was prepared and tested under different conditions, so direct comparison of adsorption capacities may not fully reflect the actual effectiveness of each material.

#### 3.2.9. Reusability

Reusability tests demonstrate considerable potential (Figure 7d). In the first cycle, the adsorption capacity (*q*_e_) for CV at *C*_o_ of 50 mg L^−1^ was 47.4 mg g^−1^, with removal efficiency (R%) of 94.9%, highlighting its high initial performance. However, as the number of cycles increased, both adsorption capacity and removal efficiency exhibited a gradual decline. By the fourth cycle, the adsorption capacity reduced to 13.2 mg g^−1^, and the removal efficiency decreased to 26.3%. After regeneration, there was a partial recovery in adsorption capacity, with *q*_e_ increasing to 39.4 mg g^−1^ and removal efficiency reaching 78.7%. Though full recovery of adsorption capacity for BHC-800 was not achieved, these results indicate that this material still shows a significant performance after regeneration.

#### 3.2.10. Adsorption Mechanisms Summary

The adsorption of crystal violet on BHC-800 is highly pH-dependent. In good agreement with the measured pH_PZC_ (Figure 6a), electrostatic attraction becomes significant at pH > 7.6, where the biochar surface acquires a negative charge, favoring cationic dye uptake. The micro-mesoporous structure of BHC-800 (pore sizes ~1.9 nm and 5.7 nm, Figure 4b) facilitates CV adsorption (molecular size: 1.28–1.34 nm) via pore filling, though this is not the dominant mechanism. The van der Waals forces also contribute, as evidenced in previous studies [24,60]. However, the removal process of CV may be supported by other mechanisms. Hence, π-π interactions play a key role due to the aromatic structure of CV and graphitic domains of BHC-800. FTIR analysis reveals intensity changes at 1600 cm^−1^ (C=C) and 560 cm^−1^ (C–H) (Figure 2b), confirming π-π stacking [24,61]. Hydrogen bonding is also plausible, as shifts in the –OH band (3300 cm^−1^) and new peaks (919 cm^−1^, 1105 cm^−1^) suggest interactions between the –OH/–PO_4_^3−^ groups of BHC-800 and nitrogen moieties of CV dye [24,28,61]. Additionally, the presence of hydroxyapatite in BHC-800 may enable ion exchange, where Ca^2+^ interacts with CV^+^, while phosphate groups (–PO_4_^3−^) could form complexes with the dye [24,28]. These findings are supported by Section 3.2.1 and Section 3.2.3. In summary, CV adsorption on BHC-800 may be governed by π-π interactions, pore filling, and hydrogen bonding, with secondary roles from electrostatic attraction and van der Waals forces.

### 3.3. Two Stages CV Dye Removal Process

In this study, new removal experiments for CV dye were conducted. In the subsequent two-stage treatment process, the removal efficiency of CV dye was evaluated across a range of initial concentrations (*C*_o_) from 100 to 1000 mg L^−1^ (Figure 8). During the preliminary salting-out step (1) using 60% brine discharge, a substantial reduction in CV concentration was achieved, with removal efficiencies ranging from 90.3% to 81.5% depending on the initial *C*_o_ values. This corresponded to residual CV concentrations of 9.7 mg L^−1^ and 185 mg L^−1^, respectively. However, complete removal of CV (100%) was not attained through this method alone, necessitating further enhancement via adsorption using the BHC-800 biochar.

As discussed before in Section 3.2.8, the adsorption of CV onto BHC-800 alone demonstrated high efficiency at lower initial concentrations (*C*_o_ ≤ 100 mg L^−1^), with removal efficiencies (R%) ranging from 73.3% to 99.4%. However, performance declined at higher concentrations, reaching only 33.8% removal at *C*_o_ = 600 mg L^−1^. In essence, when adsorption (step 2) using BHC-800 was applied following the salting-out step (1), the integrated two-stage process achieved remarkable removal efficiencies between 99.8% (for high *C*_o_ values) and 100.0% (for low *C*_o_ values), reducing CV concentrations to as low as 0.05 mg L^−1^ and 2.2 mg L^−1^, respectively (Figure 8). In other words, BHC-800 effectively adsorbed the low concentration of the residual CV remaining in the solution after the salting-out process, being its removal efficiencies R(%) > 99%.

The enhanced CV dye adsorption onto BHC-800 in the two-stage treatment, compared to the individual adsorption mode (using distilled water, Section 3.2.8), may be attributed to surface modification of the adsorbent induced by the brine solution which reinforced the removal capacity of the adsorbent. However, it should be noted that the individual experiments were not all performed under the same conditions as the two-stage treatment, which limits a direct quantitative comparison of the synergistic effect. Nonetheless, this two-stage approach significantly improved the overall removal, particularly at higher initial dye concentrations. Therefore, integrating salting-out with adsorption represents a highly effective strategy for CV removal and offers a promising solution for the treatment of water polluted by high-strength dyes.

### 3.4. Economic Efficiency of CV Removal Operation

The economic feasibility of CV dye removal is significantly influenced by costs related to raw materials, chemicals, labor, transportation, electricity, and incidental expenses. A direct operating cost method was employed to estimate brine discharge usage and biochar production expenses, focusing on laboratory-scale operations [15]. Costs are expressed in Algerian Dinar (DZD), with 1 US$ equivalent to 134.5 ± 2.0 DZD (from July 2024 to May 2025). This study utilized brine discharge (BD) from the CEO-ENASEL facility and biochar derived from bone biomass obtained free of charge from local abattoirs. Table 5 shows the details of cost calculations. The brine discharge cost primarily included transportation expenses, with 3000 L costing 1500 DZD (0.5 DZD L^−1^). Using 60% BD achieved CV dye removal efficiencies of 82–100% for initial concentrations of 0–1000 mg L^−1^. Treating 1000 L of CV dye with 60% BD required adding 300 L of brine effluent, costing 150 DZD (Laboratory scale). For industrial scales, transporting 10,000 L of BD costs 2000–4000 DZD, reducing operational costs to 60–120 DZD.

Biochar production followed a simple, chemical-free process with minimal labor. BHC-800 achieved a high yield of 63.83%, meaning 1.58 kg of feedstock was required to produce 1 kg of biochar. The cost of biochar production (C_BHC-800_) was primarily driven by electricity consumption for washing, grinding, pyrolysis, and drying. According to Panwar et al. [62], energy consumption for biochar pyrolysis typically ranges between 0.22 and 4.44 kWh kg^−1^, while in this study, the electricity usage for pyrolysis was 5.01 kWh kg^−1^. Consequently, the total production cost was estimated at 10.28–24.29 DZD kg^−1^ (Table 5). In essence, the cost per gram of CV dye adsorbed (C, DZD g^−1^ CV) was calculated as follows [15]:$$C\;(\mathrm{DZD}\cdot g^{-1}\;\mathrm{CV})\;=\frac{C_{\mathrm{BHC}‐800}\;(\mathrm{DZD}\cdot {\mathrm{kg}}^{-1})}{Q_{\max}(g\cdot {\mathrm{kg}}^{-1})}$$
where *Q*_max_ is CV-adsorption capacity of BHC-800 at ambient temperature (*T* = 20 °C).

The removal cost of 1 g of CV dye from water using the BHC-800 adsorbent is remarkably low, ranging from 0.10 to 0.23 DZD. In comparison, the cost of activated carbon is reported to be ten times higher than that of biochar [62]. Consequently, the recyclable BHC-800 adsorbent presents itself as a promising and cost-effective material for treating large volumes of CV dye effluents, particularly when combined with the salting-out process using BD.

## 4. Conclusions and Recommendations

This study explored the salting-out procedure as a preliminary step of an adsorption process and the optimized combination of both stages for an effective crystal violet removal. The research highlights the significance of leveraging saline effluents and biochar as cost-effective and eco-friendly approaches for wastewater treatment. The salting-out process, driven by high salt concentrations and pH levels, significantly enhanced the dye removal efficiency. The adsorption process using the eco-friendly BHC-800 was found to be effective, with the Freundlich isotherm and mixed adsorption mechanisms playing key roles. The Langmuir adsorption capacity of BHC-800 was 106.1 mg g^−1^ at 20 °C ± 2. The combined salting-out and adsorption approach showed superior performance, particularly at higher CV concentrations, demonstrating enhanced removal efficiency of up to ~100%. Moreover, the cost analysis revealed that removing 1 g of crystal violet using biochar (64% yield) or treating 1 m^3^ of dye-containing water with 60% brine represented less than 0.0018 and 0.9 US$, respectively. These findings offer valuable insights for sustainable wastewater management practices.

Future work should focus on optimizing this combined approach for various wastewater types, assessing environmental impacts such as effluent salinity, and exploring cost-effective strategies for large-scale application. Additionally, deeper investigation of adsorption mechanisms and potential post-treatment options will support broader implementation and ensure environmental safety.

## Figures and Tables

**Figure 1 toxics-13-01039-f001:**
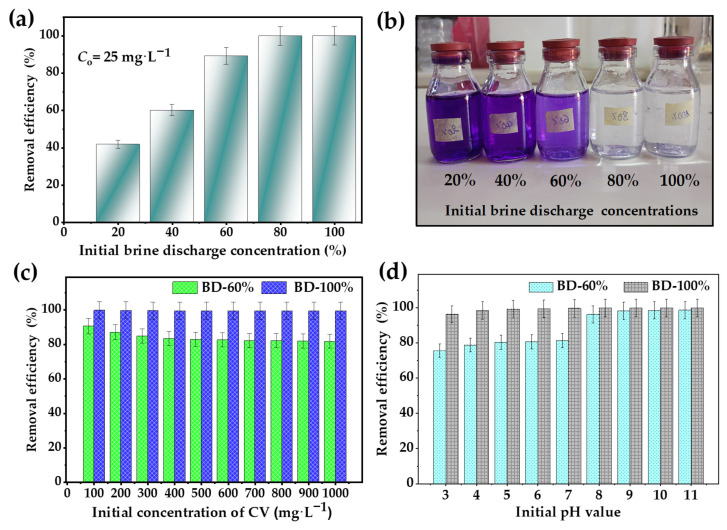
Removal efficiency of CV dye through salting out: (**a**) effect of initial concentration of brine discharge on removing 25 mg L^−1^, (**b**) digital images of removed 25 mg L^−1^ by different initial concentration of brine discharge (%), (**c**) effect of the CV initial concentration on salting out treatment, and (**d**) impact of pH on the salting-out of CV on removing 50 mg L^−1^.

**Figure 2 toxics-13-01039-f002:**
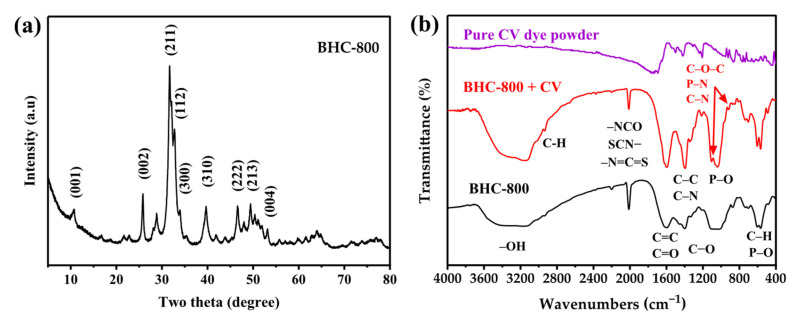
X-ray diffraction patterns of BHC-800 (**a**) and FTIR analysis before and after CV dye adsorption (**b**).

**Figure 3 toxics-13-01039-f003:**
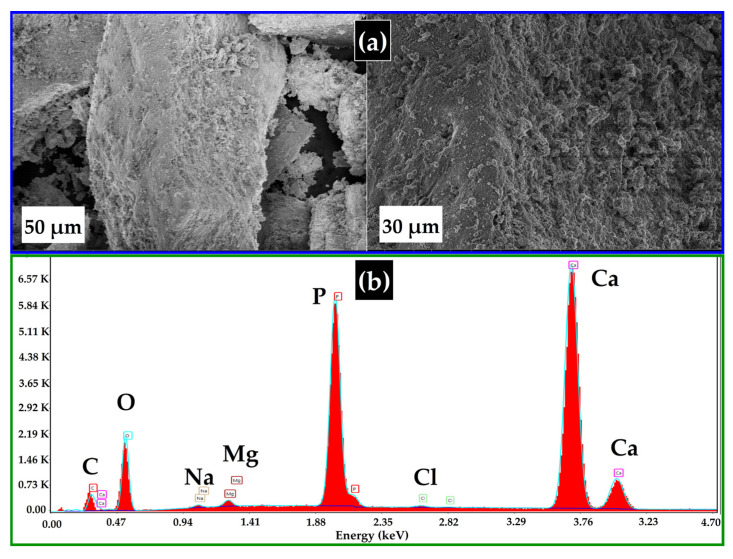
SEM micrographs (**a**) and EDX analysis (**b**) of BHC-800.

**Figure 4 toxics-13-01039-f004:**
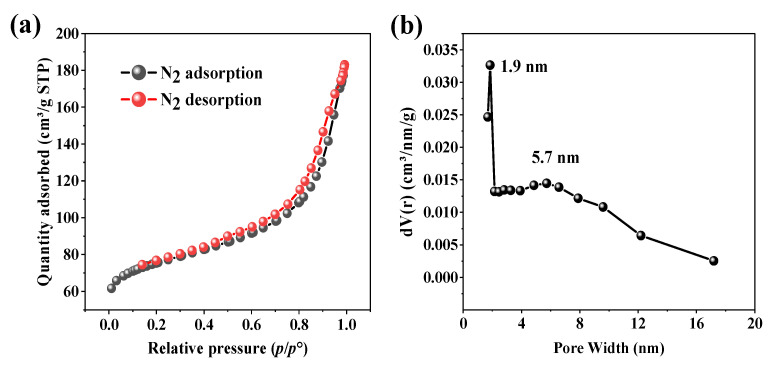
N_2_ physisorption isotherm (**a**) and pore distribution (**b**) of BHC-800. In the former, black and red dots correspond to adsorption and desorption branches, respectively.

**Figure 5 toxics-13-01039-f005:**
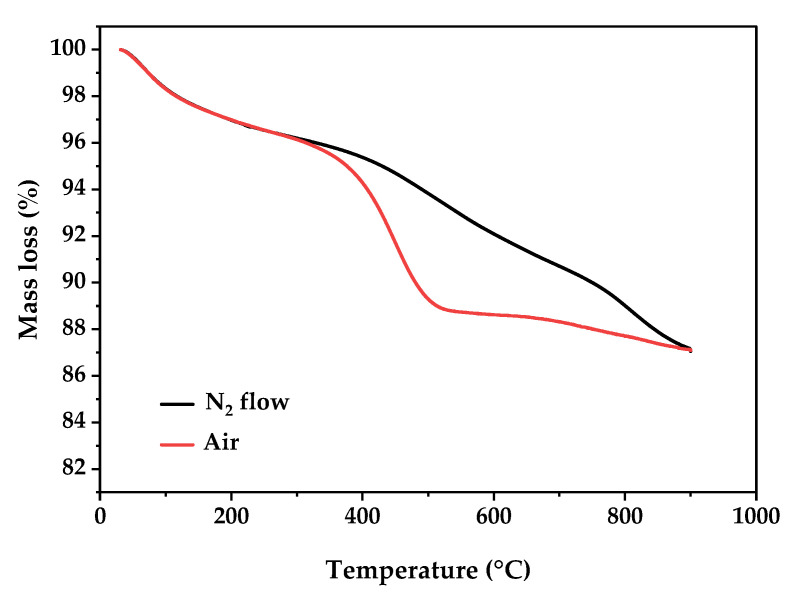
TGA curves under N_2_ flow and air of BHC-800 sample.

**Figure 6 toxics-13-01039-f006:**
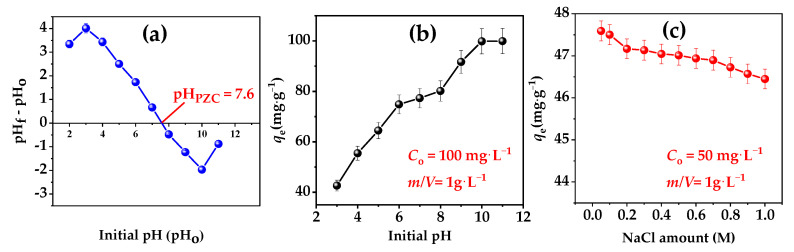
pH_PZC_ of BHC-800 (**a**), effect of pH of the medium (**b**) and ionic strength (**c**) on CV adsorption.

**Figure 7 toxics-13-01039-f007:**
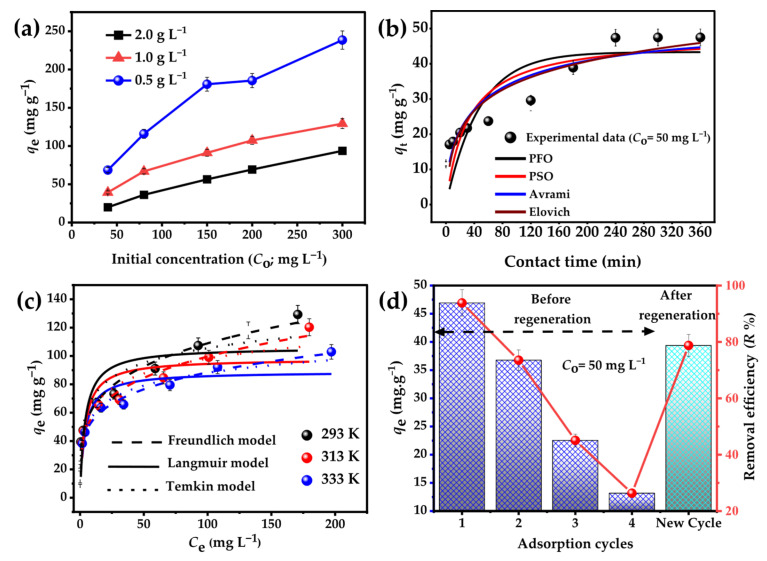
(**a**) Effect of adsorbent dose and (**b**) contact time, (**c**) isotherm curves, and (**d**) reusability for the adsorption of CV dye onto BHC-800.

**Figure 8 toxics-13-01039-f008:**
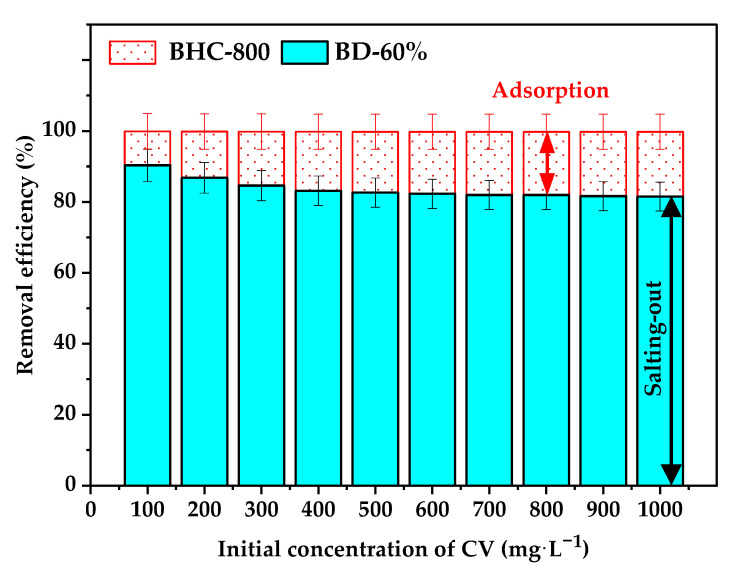
Removal efficiency of CV through successive salting-out (BD-60%) and adsorption (by BHC-800) processes.

**Table 1 toxics-13-01039-t001:** Characteristics and compositions of raw salt and brine discharge.

Parameters	Raw Salt (wt.%)	Brine Discharge (g L^−1^)
Humidity	3.23	—
Insoluble residue	0.16	—
CaSO_4_	0.69	—
MgSO_4_	2.20	—
MgCl_2_	0.30	—
NaCl	96.60	175.43
Ca^2+^	—	0.80
Mg^2+^	—	8.80
SO_4_^2−^	—	53.44
Cl^−^	—	157.40
HCO_3_^−^	—	0.90
Density	—	22 °Bé
pH	—	7.2

**Table 2 toxics-13-01039-t002:** Kinetic model constants and their respective values for the adsorption of crystal violet onto BHC-800.

Kinetic Model	Parameters	BHC-800
Experiment	*q*_e_ (mg g^−1^)	47.5
PFO	*k*_1_ (min^−1^)	0.021 ± 0.008
	*q*_e_ (mg g^−1^)	43.3 ± 4.1
	*R* ^2^	0.65
	SD	8.03
PSO	*k*_2_ (g mg^−1^ min^−1^)	6.667 × 10^−4^ ± 3.262 × 10^−4^
	*q*_e_ (mg g^−1^)	48.1 ± 4.8
	*R* ^2^	0.77
	SD	0.66
Avrami	*k*_av_ (min^−1^)	0.0157
	*q*_e_ (mg g^−1^)	49.2 ± 2.41
	*n* _av_	0.5003
	*R* ^2^	0.88
	SD	4.83
Elovich	*a*	4.143 ± 2.011
	*β*	0.111 ± 0.018
	*R* ^2^	0.88
	SD	4.70

**Table 3 toxics-13-01039-t003:** Isotherm and thermodynamic parameters for CV dye adsorption onto BHC-800.

Model	Parameter	BHC-800		
		20 °C	40 °C	60 °C
Langmuir	*K*_L_ (min^−1^)	0.256	0.263	0.267
	*Q*_max_ (mg g^−1^)	106.1	97.9	88.9
	*R* ^2^	0.68	0.73	0.71
	SD	19.97	16.22	11.74
Freundlich	*K*_F_ (min^−1^)	36.859	33.548	33.821
	*n* _F_	4.249	4.232	4.793
	*R* ^2^	0.97	0.97	0.99
	SD	5.34	5.09	3.10
Temkin	*A*	10.992	6.326	8.106
	*B*	15.072	15.211	13.155
	*R* ^2^	0.90	0.920	0.957
	SD	11.14	8.87	5.28
Thermodynamic calculations	Δ*G°* (kJ mol^−1^)	−28.3	−30.0	−32.1
	Δ*H°* (kJ mol^−1^)	−0.86	—	—
	Δ*S°* (J mol^−1^ K^−1^)	93.53	—	—

**Table 4 toxics-13-01039-t004:** Literature-based comparison of adsorption capacities of various adsorbents for CV removal vs. the results of this study.

Biochar Adsorbent	Experimental Conditions						Reference
	*m*/*V* (g L^−1^)	pH	*C*_o_ (mg L^−1^)	*T* (K)	*t* (min)	*Q*_max_ (mg g^−1^)	
**1. Found in literature**							
Palm kernel shell	16.66	—	50–500	298	24 h	24.45	[50]
Chinar leaf biochar	2.5	6.5	5–40	298	30	30.01	[51]
Rumex acetosella leaves	1	7	10–50	298	45	434.8	[52]
Durio zibethinus seed	0.8	9.9	25–50	308	25	158	[53]
Red seaweeds	1	3	5–40	298	180	5.714	[54]
Sugarcane Bagasse (650 °C)	10	7	5–30	298	20	2.94	[55]
Azadirachta indica Sawdust	0.4	7	25–100	303	90	270.27	[56]
Momordica cochinchinensis Spreng peel (BCMC550)	0.5	—	200–600	303	135	909.1	[57]
Washingtonia palm stems (BCW)	1	6	5–400	303	240	93	[58]
Golden shower pods	1	7	50–500	303	60	208.86	[59]
**2. This study**							
Inner bony core of calves’ horns	1	6.5	40–300	293	240	106.1	
	1	6.5	40–300	313	240	97.9	
	1	6.5	40–300	333	240	88.9	

**Table 5 toxics-13-01039-t005:** Cost calculations of brine discharge usage for treating 1000 L CV dye and BHC-800 production.

	Amount	Price (DZD Unit^−1^)	Total Price
**1. Brine discharge (BD) ^1^**			**60–150 (DZD)**
BD Transportation			
Laboratory scale	300 L	0.5	150 (DZD)
Industrial scale	300 L	0.2–0.4	60–120 (DZD)
**2. BHC-800 ^2^**			**10.28–24.29 (DZD kg^−1^)**
Electricity			
Production operations	0.8 kWh	1.77–4.18	1.476–3.344 (DZD kW h^−1^)
Pyrolysis	5.01 kWh	1.77–4.18	8.867–20.942 (DZD kW h^−1^)

^1^ The total cost of BD usage was calculated based on the treatment of 1000 L of CV dye. For laboratory-scale operations, transporting volumes ≤3000 L costs 1500 DZD, whereas for industrial-scale volumes ≤10,000 L, the cost ranges between 2000 and 4000 DZD. ^2^ The total cost for producing 1 kg of BHC-800 adsorbent. Production operations include washing, grinding, and drying. The average exchange rate was 1.00 US$ = 134.5 ± 2.0 DZD from July 2024 to May 2025.

## Data Availability

The data that support the findings of this study are available from the corresponding author upon reasonable request.

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
