# Peer review of "Ultra-Efficient Removal of Crystal Violet Dye Using Industrial Brine and Horn-Derived Biochar: Synergistic Action of Salting-Out/Adsorption"

_toxics, 2025, doi:10.3390/toxics13121039_

Round 1

Reviewer 1 Report

Comments and Suggestions for Authors

Dear authors, 

please see the attached file.

Reviewer 2 Report

Comments and Suggestions for Authors

The study presents a relevant and well-conducted experimental approach with promising results regarding dye removal using biochar under saline conditions. The topic is timely and environmentally meaningful, and the manuscript demonstrates solid technical work and careful experimentation.

However, several aspects related to the depth of discussion, data presentation, and integration of characterization with kinetic and thermodynamic results require improvement to enhance the scientific clarity and impact of the paper.

In the attached document, I provide detailed and constructive comments intended to help the authors strengthen their manuscript and make full use of the valuable data they have generated. I sincerely encourage the authors to consider these suggestions carefully, as the work has significant potential for publication once these revisions are addressed.

Round 2

Reviewer 1 Report

Comments and Suggestions for Authors

Dear authors,

Please consider the following suggestions:

Line 134 and 138 – why sating-out process take place at 15ºC and adsorption on biochar take place at 20ºC?

I suggest the inclusion of the clarification sent as a reply to me in the manuscript.

Line 268 – now 284 -  “This porous structure is particularly advantageous for effectively capturing crystal violet dye during the adsorption process.”

 I suggest the removal of this sentence from the manuscript.

Reviewer 2 Report

Comments and Suggestions for Authors

I would like to acknowledge the substantial improvements made in the revised version of your manuscript. Many sections—particularly the characterization and the organization of the kinetic and isotherm models—are now clearer and better structured. Your study provides relevant insights into the combined effect of brine and biochar on dye removal, and your experimental effort is appreciated.

However, after carefully reviewing the revised manuscript and supplementary material, several technical issues remain that require clarification before the work can be accepted. These points are critical because they directly affect the interpretation of your kinetic and thermodynamic results:

(1) Avrami kinetic model selection is not justified.
Although the manuscript states that the Avrami model is the best fit, your calculated qe (2064.7 mg/g) deviates substantially from the experimental data presented in Fig. 7b. In contrast, the pseudo-first-order (PFO) and pseudo-second-order (PSO) models visually align more closely with the experimental qe values. This discrepancy must be addressed, as it calls into question the validity of identifying Avrami as the “best-fitting” model.

(2) Thermodynamic parameters require reevaluation.
The reported value of enthalpy change (ΔH°) is unusually low and should be rechecked. Given the adsorption mechanism proposed and the ionic environment of brine, a higher magnitude would typically be expected. Please verify both the slope of the Van’t Hoff plot and the definition of the equilibrium constant used (Kc). This is especially important because the Kc values do not appear to arise from a well-demonstrated equilibrium state.

(3) The equilibrium between dye and brine requires supporting data.
Although the manuscript states that equilibrium was reached prior to adding the biochar, no experimental curve (C/C₀ vs t) is presented to support this claim. Without visual confirmation of this equilibrium, the thermodynamic calculations may not accurately represent the system.

(4) Comparability among the three systems is improved but still not entirely clear.
You indeed present results for (i) dye + biochar, (ii) dye + brine, and (iii) dye + brine + biochar. However, not all experiments are performed under comparable conditions, which prevents a direct quantitative assessment of synergistic behavior. This does not invalidate your conclusions but should be clarified in the text so that the reader understands the limitations.

Additionally, some transitions and explanations in the methodology remain difficult to follow because key information is separated across sections. A more linear description of procedures would improve clarity and reproducibility.

Overall, the manuscript is significantly improved, but these technical issues—especially those related to kinetic model validity and thermodynamic consistency—must be resolved to strengthen the scientific reliability of your conclusions. I encourage you to revise these aspects carefully.
